# Multiomics Study of a Novel Naturally Derived Small Molecule, NSC772864, as a Potential Inhibitor of Proto-Oncogenes Regulating Cell Cycle Progression in Colorectal Cancer

**DOI:** 10.3390/cells12020340

**Published:** 2023-01-16

**Authors:** Ntlotlang Mokgautsi, Yu-Cheng Kuo, Chien-Hsin Chen, Yan-Jiun Huang, Alexander T. H. Wu, Hsu-Shan Huang

**Affiliations:** 1Ph.D. Program for Cancer Molecular Biology and Drug Discovery, College of Medical Science and Technology, Taipei Medical University and Academia Sinica, Taipei 11031, Taiwan; 2Graduate Institute for Cancer Biology & Drug Discovery, College of Medical Science and Technology, Taipei Medical University, Taipei 11031, Taiwan; 3Department of Pharmacology, School of Medicine, College of Medicine, Taipei Medical University, Taipei 11031, Taiwan; 4School of Post-baccalaureate Chinese Medicine, College of Chinese Medicine, China Medical University, Taichung 40402, Taiwan; 5Department of Colorectal Surgery, Wan Fang Hospital, Taipei Medical University, Taipei 11031, Taiwan; 6Department of Surgery, College of Medicine, Taipei Medical University, Taipei 11031, Taiwan; 7Division of Colorectal Surgery, Department of Surgery, Taipei Medical University Hospital, Taipei Medical University, Taipei 11031, Taiwan; 8The Ph.D. Program of Translational Medicine, College of Medical Science and Technology, Taipei Medical University, Taipei 11031, Taiwan; 9Clinical Research Center, Taipei Medical University Hospital, Taipei Medical University, Taipei 11031, Taiwan; 10T.M.U. Research Center of Cancer Translational Medicine, Taipei Medical University, Taipei 11031, Taiwan; 11Graduate Institute of Medical Sciences, National Defense Medical Center, Taipei 11490, Taiwan; 12School of Pharmacy, National Defense Medical Center, Taipei 11490, Taiwan; 13Ph.D. Program in Drug Discovery and Development Industry, College of Pharmacy, Taipei Medical University, Taipei 11031, Taiwan

**Keywords:** colorectal cancer, drug resistance, protein–ligand interaction, molecular docking simulation, small molecule

## Abstract

Colorectal cancer (CRC) is one of the most prevalent malignant tumors, and it contributes to high numbers of deaths globally. Although advances in understanding CRC molecular mechanisms have shed significant light on its pathogenicity, current treatment options, including combined chemotherapy and molecular-targeted agents, are still limited due to resistance, with almost 25% of patients developing distant metastasis. Therefore, identifying novel biomarkers for early diagnosis is crucial, as they will also influence strategies for new targeted therapies. The proto-oncogene, *c-Met*, a tyrosine kinase that promotes cell proliferation, motility, and invasion; *c-MYC*, a transcription factor associated with the modulation of the cell cycle, proliferation, apoptosis; and cyclin D1 (*CCND1*), an essential regulatory protein in the cell cycle, all play crucial roles in cancer progression. In the present study, we explored computational simulations through bioinformatics analysis and identified the overexpression of *c-Met/GSK3β/MYC/CCND1* oncogenic signatures that were associated with cancer progression, drug resistance, metastasis, and poor clinical outcomes in CRC. We further demonstrated the anticancer activities of our newly synthesized quinoline-derived compound, NSC772864, against panels of the National Cancer Institute’s human CRC cell lines. The compound exhibited cytotoxic activities against various CRC cell lines. Using target prediction tools, we found that *c-Met/GSK3β/MYC/CCND1* were target genes for the NSC772864 compound. Subsequently, we performed in silico molecular docking to investigate protein–ligand interactions and discovered that NSC772864 exhibited higher binding affinities with these oncogenes compared to FDA-approved drugs. These findings strongly suggest that NSC772864 is a novel and potential antiCRC agent.

## 1. Introduction

Colorectal cancer (CRC) is the third most common malignancy worldwide and ranks as the fourth leading cause of cancer deaths [1,2]. Although remarkable progress has been achieved through current multimodality therapies, including chemotherapy and molecular-targeted agents, such as cetuximab [3,4,5,6,7], most patients eventually still become resistant to therapy [8]. Approximately 25% of CRC patients develop advanced metastatic cancer, which leads to poor clinical outcomes [9]. The 5-year overall survival rate in the initial stage is approximately 90%; however, survival of advanced-stage CRC patients is merely 10% [10]. Dysregulation of various genetic and molecular processes was reported in CRC, thus necessitating the urgent need to identify novel biomarkers, which can be used for early diagnosis and drug development [11,12]. The mesenchymal–epithelial transition (c-Met) is a tyrosine kinase receptor in the membrane encoded by the *c-Met* proto-oncogene [13]. The dysregulation of *c-Met* signaling was reported in CRC patients, and it promotes tumor angiogenesis, growth, and metastasis [14,15,16,17,18,19]. Studies showed that *c-Met* overexpression at the messenger (m) RNA and protein levels is linked to CRC progression, stemness, distant metastasis, and poor clinical outcomes [20,21,22].

Moreover, recent reports demonstrated that when activated in CRC patients, *c-Met* promotes resistance to cetuximab [23]. Therefore, elucidating the molecular mechanisms of c-Met overexpression in CRC pathogenesis would be of great significance for determining an effective approach for CRC therapy. Moreover, *c-Met*-dependent and -independent signaling contributes to *Met* receptor-mediated cell cycle progression [20]. The c-Met pathway has been shown to modulate various downstream signaling pathways in tumors, including PI3K/AKT and Wnt/β-catenin, among others [24,25]. The association of c-MET with Wnt/β-catenin in particular has been demonstrated to promote cancer proliferation, progression, and metastasis [26]. Several other studies have also shown that c-Met expression enhances the Wnt/β-catenin signaling and impedes GSK3β from phosphorylating β-catenin in CRC; this subsequently promotes β-catenin translocation into the nucleus leading to cancer initiation [27,28]. It has been shown that the inhibition of c-Met also suppresses Wnt pathway signaling transduction activities in cancer cells [29,30]. Glycogen synthase kinase 3 (GSK3), a serine/threonine kinase, and its expression in CRC associates with the regulation of cell cycle, cancer proliferation, and apoptosis [29], thus making it a potential biomarker for therapeutic development in CRC [31,32,33]. CRC is mainly known for its heterogeneity, including alterations in the expression of *c-MYC*, which was shown to be associated with CRC progression through the *c-Met* signaling pathway [34]. Accumulating studies have shown that MYC is a potential target of β-catenin signaling in CRC cells [35]. The *c-MYC* proto-oncogene is a transcription factor that modulates the cell cycle, proliferation, and apoptosis. Additionally [36,37], when upregulated in CRC, it is associated with cancer progression, recurrence, and metastasis [38,39,40,41,42,43,44]. The overexpression of *c-MYC* was shown to enhance the expression of cyclin D1 (*CCND1*) through activating beta-catenin [45], thus promoting CRC proliferation [46]. *CCND1* is a crucial regulator of cell progression through the G_1_ to S phases of the cell cycle via binding to cyclin-dependent kinase 4 (*CDK4*) and *CDK6* [47].

Studies showed that *CCND1* overexpression is linked to poor prognoses in CRC patients [48,49]. These findings suggest crosstalk among c-Met/GSK3β/MYC/CCND1 as a potential prognostic biomarker of CRC, which might be used to develop novel and effective therapeutics. In this study, we utilized a bioinformatics analysis to identify potential oncogenes, which are overexpressed in CRC Interestingly, our results reveal that *c-Met/GSK3β/MYC/CCND1* were among the top 25 differentially expressed genes (DEGs) in CRC, extracted from the GEO microarray datasets, GSE4107 and GSE41328. MicroRNAs (miRNAs) are noncoding short RNAs (21–25 nucleotides) that negatively regulate gene expression at the translational level by binding to 3′UTR (untranslated region) of their target mRNAs leading to either their degradation or translational inhibition [50,51]. The dysregulation of expression levels in various miRNAs, including miR-26a, has been shown to associate with proliferation and invasion in different cancer types, including CRC [52]. miR-26a is significantly overexpressed in CRC. However, its function and clinical significance in CRC remain elusive [53]. Through computational simulation, we identified miR-26a bound to the 3′UTR of *GSK3β/MYC/CCND1*, thus suggesting a close association of *Met/GSK3β/MYC/CCND1* and miR-26a promoting cancer proliferation, invasion, migration, drug resistance, and stemness metastasis in CRC patients.

The search for a novel and effective cancer therapy has increased over the years due to resistance to current treatment modalities. Quinoline is a heterocyclic aromatic nitrogen compound commonly used as a parental compound to synthesize various molecules [54]. Numerous studies demonstrated that several quinoline-derived molecules exhibit anticancer activities due to their ability to inhibit specific proteins [55]. One of the quinoline alkaloids, camptothecin, is an S-phase cell cycle-specific anticancer agent shown to induce topoisomerase I-dependent DNA breaks [56,57]. Moreover, sunitinib, a multitargeted receptor tyrosine kinase (RTK) inhibitor [58], which was approved as an anticancer agent in gastrointestinal stromal tumor patients, was shown to inhibit cancer cell growth expressing RTKs [59,60]. In the current study, we evaluated the anticancer effects of NSC772864, a novel small-molecule derivative of quinoline and sunitinib, which was recently developed in our lab [61,62,63,64,65,66]. Computational simulations are extensively used in drug discovery and development; therefore, we applied in silico predictions and screened our NSC772864 small molecule for its anticancer activities in a panel of National Cancer Institute (NCI) CRC cell lines to predict novel drug targets [67]. We further identified c-Met/GSK3β/MYC/CCND1 as druggable candidates for this compound and further performed molecular docking simulations of ligand–receptor interactions of NSC772864 in complex with c-Met/c-MYC/CCND1 study summary is shown in Figure 1.

## 2. Material and Methods

### 2.1. Retrieval of the Top 25 DEGs in CRC

The top overexpressed DEGs in colon adenocarcinoma (COAD) samples were retrieved from the following databases: NCBI GEO two datasets, GEO microarray datasets, and GSE4107 and GSE41328 (https://www.ncbi.nlm.nih.gov/ 21 June 2022). We used ULCAN, a web-based tool to analyze The Cancer Genome Atlas (TCGA) level 3 RNA sequencing (RNA-Seq) and clinical data from tumor tissues compared to adjacent normal tissues [68]. The top overexpressed genes with significantly different transcripts per million (TPM) values (*p* < 0.001) were first selected and represented in a heatmap.

### 2.2. DEG Validation

To validate expression levels of the identified DEGs, including *c-c-Met/GSK3β/MYC/CCND1* in CRC, we explored the tumor, normal, and metastatic plot tool (https://tnmplot.com/analysis/ 28 June 2022), which compares selected genes from RNA-Seq-based data [69]. A comparative analysis was performed using the Kruskal–Wallis test method, with *p* < 0.05 considered statistically significant. Expressions of *c-Met/GSK3β/MYC/CCND1* based on individual cancer stages were analyzed by utilizing ULCAN software. Furthermore, we used GECO (https://proteinguru1.shinyapps.io/geco/ 28 June 2022), a gene expression correlation analytical tool, which differentiates two expression profiles into positive and negative correlations [70], with positive Pearson correlation coefficients and *p* < 0.05 as statistically significant.

### 2.3. Interacting Network Construction and Functional Enrichment Analysis

A protein interaction network (PIN) was analyzed using STRING, a protein–protein interaction (PPI) and functional enrichment analysis web tool (https://string-db.org/ 29 June 2022) [71]. We further downloaded functional enrichment clusters, from the STRING analysis, and these included Kyoto Encyclopedia of Genes and Genomes (KEGG) pathways and gene ontology (GO), which involves biological processes (BPs), molecular functions (MFs), and cellular components (CCs). Functional enrichments of the selected clustering network were further classified using the Database for Annotation, Visualization, and Integrated Discovery (DAVID) (https://david.ncifcrf.gov/ 29 June 2022) functional annotation tool [72]. Moreover, we explored the functional enrichment software database, FunRich (http://www.funrich.org/ 29 June 2022), to further analyze interactions within the networks, with *p* < 0.05 considered significant.

### 2.4. Analysis of c-Met/GSK3β/MYC/CCND1 Genetic Alterations and Mutation Analysis in CRC Tissues

To determine relationships between *c-Met/GSK3β/MYC/CCND1* mutations and alterations in gene expressions, we utilized the muTarget platform (http://www.mutarget.com 06 August 2022) and web tool that link gene expression alterations and mutation statuses in cancers [73]. We further used the cBio Cancer Genomics Portal (cBioPortal; http://cbioportal.org 6 August 2022), an online interactive platform for translational research that provides analysis of cancer genomic datasets [74], to determine alteration frequencies of selected gene signatures in CRC tissues. Furthermore, based on the genetic alteration profile of the cohorts, each was categorized into one of two groups: an unaltered group and an altered group. The analytical results were considered significant at *p* < 0.05.

### 2.5. Drug Sensitivity and Gene Expression Profiling for c-Met/GSK3β/MYC/CCND1 Oncogenes in Colon Tumors

Correlations between expressions of *c-Met/GSK3β/MYC/CCND1* oncogenes and drug sensitivity were determined using the GSCALite database, an online tool for Gene Set Cancer Analysis (GSCA) (http://bioinfo.life.hust.edu.cn/web/GSCALite/ 11 August 2022) [75]. The analysis was performed using the top 38 drugs in a pancancer database. Expression levels of individual genes in the gene set were analyzed using a Spearman correlation analysis in response to the drug sensitivity (50% inhibitory concentration (IC_50_)) of small molecules. Additionally, correlations considered statistically significant displayed a false discovery rate (F.D.R.) of <0.05.

### 2.6. c-Met, MYC, and CCND1 Are Potential Drug Targets of NSC772864

The Developmental Therapeutics Program (D.T.P.)-COMPARE online platform, from the NCI (https://dtp.cancer.gov/ 15 June 2022) is an online pattern recognition algorithm used to predict molecular targets based on ARRAYCGH_GRAY and detect a compound’s similarities to NCI synthetic compounds and standard agents. Herein, we used 50% growth inhibition (GI_50_), which is an endpoint of the IC_50_, and the National Safety Code (N.S.C.) number (772864) as delimiters [76]. Furthermore, we used the PharmMapper Server (http://59.78.96.61/pharmmapper 14 August 2022), an integrated pharmacophore mapping platform, to identify potential drug target candidates [77].

### 2.7. Evaluation of Drug-Likeness, Absorption, Distribution, Metabolism, and Excretion (ADME) Properties and Friendliness of NSC772864

Identifying novel and potential drug candidates early in drug discovery and development is crucial, as it reduces time and costs; herein, we applied the drug-likeness concept based on specific criteria [78,79]. To evaluate the drug-likeness, pharmacokinetics (PKs), and medicinal chemistry of the NSC772864 compound, we utilized swissADME (http://www.swissadme.ch/ 15 June 2022), an online platform used to evaluate PKs, drug-likeness, and the medicinal chemistry friendliness of compounds [80]. We analyzed the drug-likeness properties according to the Lipinski (Pfizer) rule of five, Ghose (Amgen), Veber (GSK), and Egan (Pharmacia) standards and further showed relationships between the PKs and physicochemical properties [81]. Moreover, we analyzed the gastrointestinal absorption (GIA) and blood–brain barrier (BBB) penetration properties using the brain or intestinal estimated permeation (BOILED-Egg) model [82]. The Abbot bioavailability score was determined based on the probability of the compound having at least 10% oral bioavailability in rats or measurable Caco-2 permeability [83].

### 2.8. In Vitro Anticancer Screening of NSC77286 against NC1 60 CRC Cells

NSC772864 was submitted to the NCI Developmental Therapeutics Program (DTP) for screening for potential antiproliferative and cytotoxic effects against a panel of NCI 60 CNS cell lines, in agreement with the protocol outlined by the NCI (https://dtp.cancer.gov/ 30 June 2022). The compound was tested at an initial dose of 10 μM and further tested for dose-dependent treatment. The results show that NSC772864 exhibited antiproliferative activities against several CNS cell lines [84].

### 2.9. Molecular Docking Analysis

A docking analysis was performed using AutoDock Vina (vers. 1.5.6; Scripps Research Institute Molecular Biology, La Jolla, CA, USA). Accordingly, crystal structures of the *c-Met* (PDB:3VW8), *MYC* (PDB:6G6K), and *cyclin D1* (PDB:2W9Z) proto-oncogenes were retrieved from the Protein Data Bank (PDB) (https://www.rcsb.org/ 26 July 2022). Furthermore, the three-dimensional (3D) structure of NSC772864 was assembled with the Avogadro molecular visualization tool [85]. Moreover, the 3D structures of the standard inhibitors crizotinib (CID:11626560), alobresib (CID:86281210), and trilacoclib (CID:68029831) were downloaded from PubChem as SDF files, and the files were further converted to PDB format using PyMol software [86]. All the PDB format files were subsequently converted into PDBQT file format and processed for docking using autodock software. Finally, the docking results were interpreted and visualized using BIOVIA discovery studio software [79,87,88].

### 2.10. Statistical Analysis

Pearson’s correlations were used to assess the correlations of *c-Met/GSK3β/MYC/CCND1* expressions in CRC cancer types.The statistical significance of DEGs was evaluated using the Wilcoxon test. * *p* < 0.05 was accepted as being statistically significant.

## 3. Results

### 3.1. High Expressions of c-Met/GSK3β/MYC/CCND1 Promote Colon Cancer Progression

The top overexpressed DEGs in COAD tissues compared to normal tissues were retrieved from two NCBI GEO datasets, GSE4107 and GSE41328 (Figure 2A,B). We used a Venn diagram to demonstrate the overlapping upregulated genes from the two datasets (Figure 2C). Subsequently, we analyzed these DEGs using the ULCAN database to construct the heatmap with the overexpressed genes. The c-Met/GSK3β/MYC/CCND1 oncogenic signature was identified among the top-ranking DEGs. The gene expression level is represented as log2(TPM +1) (Figure 2D). The TNM plots revealed a high expression of *c*-Met/GSK3β/MYC/CCND1 in tumor tissues compared to normal colon tissues (Figure 2E,F). The increased expression of the c-Met/GSK3β/MYC/CCND1 signature in different CRC stages was compared to normal tissues (Figure 2E–G), and *p* < 0.05 indicated statistical significance (Figure 2H–J).

### 3.2. PPI Network (PIN) Construction and Enrichment Analysis

PPIs were analyzed using the STRING database, and the analytical results show an average local clustering coefficient of 0.920 and interaction enrichment with *p* = 0.0135. Interactions were analyzed based on coexpression, gene fusion, and co-occurrence (Figure 3A). Functional enrichments with the clustering network were retrieved from the STRING analysis, and they included GO involving BPs, which were analyzed using FunRich software. Among the top 10 enriched BPs were E-cadherin signaling, the FOXM1 transcription factor network, G_1_ phase, cyclin D, and mitosis G_1_–G_1_/S phases (Figure 3B). For further analysis, we used the DAVID database to evaluate enriched KEGG pathways and CCs. Interestingly, among the top 15 enriched pathways were MYC, CCND1, lysine acetyltransferase 2A (KAT2A), sirtuin 1 (SIRT1), S-phase kinase-associated protein, c-Met, and hepatocyte growth factor, while the enriched CCs included CCND1, cyclin-dependent kinase 4 (CDK4), and KAT2A (Figure 3C,D), with *p* < 0.05 considered significant.

### 3.3. Crosstalk between Overexpression of Met/GSK3β/MYC/CCND1 Oncogenes and Upregulated miR-26a Are Associated with CRC Cancer Progression

Migration and metastasis play a significant role in determining the clinical outcomes of colorectal cancer (CRC). A major factor for metastasis is the acquired capacity of the cell to proliferate and invade normal tissues. Herein, we investigated the role of miR-26a in regulating CRC cell proliferation and migration, through the crosstalk with upregulated Met, GSK3β, MYC, CCND1, and oncogenic signatures [53]. We used the FunRich and NetworkAnalyst comprehensive gene and miRNA expression profiling platforms [89,90], and profiled all the miRNAs that targets c-Met, GSK3β, MYC, and CCND1 oncogenes specifically (Figure 4). For further analysis, we identified OncomiR-26a to be enriched within the same clustering network and linking with Met/GSK3β/MYC/CCND1 oncogenes (Figure 5A), and to explore further, we used the ENCORI tool and identified miR-26a to be downregulated in CRC samples as compared to normal samples (Figure 5B). Moreover, we identified the miR-26a-1-3p site in the 3′ untranslated region (UTR) of GSK3β, MYC, and CCND1 mRNA, thus predicting a link between miR-26a and Met/GSK3β/MYC/CCND1 in cancers (Figure 5C,D). To explore further, we performed an enrichment analysis and identified all the enriched gene ontology including the biological process and the affected KEGG pathways (Figure 5F,G).

### 3.4. Genomic Alterations in c-Met/GSK3β/MYC/CCND1 Signatures Are Associated with Poor Prognoses of CRC Cohorts

Mutations of the *c-Met* oncogene were first linked to changes in gene expressions in CRC at the genotypic level using the muTarget tool. Herein, high expression levels of MSANTD3 were associated with mutations on the *c-Met* gene compared to wild-type (WT) *c-Met* (Figure 6A). Moreover, changes in the expression levels of the *c-Met/GSK3β/MYC/CCND1* oncogenes to mutations of the top genes at the target level were also analyzed. Interestingly, we found that BRAF, NOTCH2, and ODF2 mutations were associated with the overexpression of *c-Met, MYC,* and *CCND1* compared to the WT group (Figure 6B–D). In a further analysis, we determined the alteration frequencies of *c-Met/GSK3β/MYC/CCND1* gene signatures in CRC, COAD, and rectal adenocarcinoma (READ) tissues. Interestingly, the *c-Met/GSK3β/MYC/CCND1* oncogenes were shown to be more mutated in CRC (Figure 6E,F). We continued to analyze the genetic alterations from sample subtypes and found that *c-Met/GSK3β/MYC/CCND1* were more altered in COAD chromosomal instability (CIN) and in microsatellite instability (MSI) and had higher numbers of mutations and amplifications in CRC tissues as shown in the bar graph (Figure 6F). We also identified all the altered and unaltered genes associated with *c-Met/GSK3β/MYC/CCND1* signatures (Figure 6G) and further determined the distributions of *c-Met/GSK3β/MYC/CCND1* mutations in CRC across the protein domains (Figure 6H–J).

### 3.5. Rational Scaffold-Hopping Protocol for the Design of NSC772684

Computer-based drug discovery tools have been widely used over the years and assist in the processes of discovering novel drugs in various ways by offering many more benefits, such as reducing the screening of multiple compounds experimentally and reducing costs [91,92]. Thus, identifying novel bioactive compounds is vital. Scaffold hopping is one of the computational identifications of compound structure–activity relationships (SARs) [93]. Quinoline-based compounds play crucial roles in anticancer drugs. Different types of anticancer molecules have been designed from quinoline derivatives and were demonstrated to exhibit antiproliferative effects in various cancer cells [94,95]. Camptothecia, known as “happy tree”, is one of the S-phase cell cycle-specific anticancer drugs derived from quinoline, which was shown to inhibit DNA topoisomerase I [96,97]. In the present study, the scaffold hopping of quinoline (C_9_H_7_N) (and the FDA-approved multitargeted receptor tyrosine kinase (RTK) inhibitor sunitinib[(Z)-N-(2-(diethylamino) ethyl)-5-((5-fluoro-2-oxoindolin-3-ylidene) methyl)-2,4 dimethyl-1H-pyrrole-3-carboxamide), which was also demonstrated to inhibit GIST [59,98,99,100,101,102], led to the discovery of NSC772864 (9-chloro-6-((2-(diethylamino) ethyl) amino)-11H-indeno[1,2-c] quinolin-11-one), a newly synthesized small molecule from our lab (Figure 7).

### 3.6. c-Met, MYC, and CCND1 Are Potential Drug Targets of NSC772864

We explored the DTP-COMPARE web server, and the analytical results show that c-Met, MYC, and CCND1 are potential targets of NSC772864. The predictions were based on an ARRAYCGH_GRAY analysis, and we further compared the compound’s similarities to NCI synthetic compounds and standard agents as shown in Table 1. In a further analysis, we used the independent tool, PharmMapper, which identifies potential drug targets using a pharmacophore-matching approach. The analysis from the PharmMapper run was sorted according to the fit score. The results reveal the top 300 target genes of NSC772864, which were ranked by normalized fit scores ranging from 0.4875 to 0.9936. Interestingly, c-Met and CCND1 were identified among the 300 target genes for NSC772864. Accordingly, the c-Met/CCND1/cMYC–NSC772864 complex exhibited three hydrophobic interactions for c-Met and CCND1 and 15 hydrophobic interactions for c-MYC, with normalized fit scores of 0.6849, 0.6942, and 0.1667 (Figure 8A–C), respectively.

### 3.7. Drug Sensitivity Analysis for the c-Met/GSK3β/MYC/CCND1 Oncogenes in Colon Tumors

We explored the GSCA to determine the drug sensitivity profile data of c-Met/GSK3β/MYC/CCND1 against different cancer cell lines. Herein, correlations of c-Met/GSK3β/MYC/CCND1 gene expressions and drug sensitivity were analyzed based on values of the area under the dose–response curve (AUC) for drugs and gene expression profiles for all cell lines. The expression of each gene in the gene set was analyzed using a Spearman correlation analysis of the drug sensitivity (IC50) of the small molecule against various cell lines in the Cancer Therapeutics Response Portal (CTRP) and Genomics of Drug Sensitivity (GDSC) databases. The correlation coefficients indicated that increased gene expression levels were resistant to the drug. Herein, high messenger (m) RNA expression levels of c-Met and CCND1 (as indicated by orange circles) were associated with drug resistance (Figure 9).

### 3.8. Evaluation of Drug Likeness, ADME Properties, and Friendliness of NSC772864

To evaluate the drug-likeness, PKs, medicinal chemistry, and ADME and toxicity (ADc-Met) properties of NSC772864, we used the swissADME online platform. Accordingly, we analyzed the drug-likeness properties of the compound according to Lipinski (rule of five), Egan, Ghose, and Veber standards. Moreover, we also showed associations of PKs with physiochemical properties. In a further analysis, we used the BBB penetration properties, using the brain or intestinal estimated permeation (BOILED-Egg) model [82,103], to predict the GIA and BBB penetration properties of NSC772864. The Abbot bioavailability score was determined based on the probability of the compound having at least 10% oral bioavailability in rats or measurable Caco-2 permeability [83] (Figure 10A,B). Drug target prediction software showed that NSC772864 has different targetable proteins, most of which were family A G protein-coupled receptors followed by kinases (Figure 10C).

### 3.9. In Vitro Anticancer Screening of SJ3 against NC1-60 CRC Cells

The NSC772864 compound was submitted to the NCI DTP for screening for potential in vitro anticancer activities against a panel of NCI CRC cell lines. NSC772864 was first tested at an initial dose of 10 μM. The compound exhibited antiproliferative and cytotoxic activities against CRC cell lines as indicated by the percentage growth altered by treatment. Accordingly, the growth inhibition percentage revealed antiproliferative effects against HCC-2988 cells of 88.65% and cytotoxic activities, showing KM12 to be more sensitive with -53.42% growth inhibition followed by HCT15 (−50 38%), COLO205 (−45.69%), SW620 (−43.72%), HCT116 (−36.96%), and HT29 cells (−9.75%) (Figure 11A-I). Because NSC772864 showed potential antiproliferative and cytotoxic activities at an initial dose of 10 μM, the compound was further tested in a dose-dependent manner. Interestingly, the results show potential antiproliferative effects in a dose-dependent manner (Figure 11J). We further investigated the in vitro GI50. The results ranged from 0.16 to 2.85 μM in colon cancer cell lines, with HCT-15 being more responsive at 0.16 μM, followed by HCT116 at 0.38 μM, SW-620 at 0.39 μM, HT-29 at 0.57 μM, KW12 at 0.94 μM, and COLO 205 at 2.15 μM, with HCC-2998 at 2.85 μM showing the least responsiveness compared to the aforementioned cell lines (Figure 11K).

### 3.10. Molecular Docking Analysis

To investigate the potential inhibitory effects of NSC772864, we conducted a molecular docking analysis. The results obtained from ligand–receptor complexes show unique different binding energies of NSC772864 with c-Met (ΔG = −8.0 kcal/mol, Figure 12A,B), GSK3β (=−8.6 kcal/mol, Figure 13A,B), MYC (ΔG = −9.1 kcal/mol, Figure 14A,B), and CCND1 (ΔG = −8.0 kcal/mol, Figure 15A,B). These results were further compared to standard inhibitors of crizotinib for c-Met (Figure 12C,D), AZD1080 for GSK3β (Figure 13C,D), alobresib for MYC. (Figure 14C,D), and trilacoclib for CCND1 (Figure 15C,D), with respective binding energies of ΔG = −8.1, −8.4, −7.6, and −7.4 kcal/mol. Interestingly, NSC772864 showed higher binding energies compared to AZD1080, alobresib, and trilacoclib; however, crizotinib was shown to possess a slightly higher binding energy compared to NSC772864. In a further analysis, we used Pymol and Discovery Studio for the analysis and visualization of the results. The NSC772864/c–Met complex showed interactions of conventional hydrogen bonds (H-bonds) with ASP1222 (2.20 Å), GLU1127 (1.85 Å), and LYS1110 (2.68 Å) compared to the crizotinib/c-Met complex, which showed H-bonds with ASP1222 (2.20 Å), GLU1258 (2.19 Å), ASP1204 (1.96 Å), and ARG1227 (2.60 Å). The NSC772864/GSK3β complex displayed H-bond interactions with ASN64 (2.56 Å), compared to the AZD1080/GSK3β complex, which displayed H-bond interactions with PHE67(2.30Å) and GLN185 (2.70 Å). The NSC772864/myc complex displayed H-bond interactions with SER221 (2.48 Å), compared to the alobresib/myc complex, which had no H-bond interactions. Moreover, the NSC772864/ccnd1 complex exhibited an H-bond interaction with ARG26 (2.88 Å), compared to trilacoclib/ccnd1, which showed H-bond interactions with HIS163 (2.29 Å) and ALA133 (2.04 Å). The interactions were sustained by van der Waals forces, carbon–hydrogen bonds, π bonds, and alkyl bonds as shown in (Table 2, Table 3 and Table 4).

## 4. Discussion

In recent years, the survival rates of CRC patients have significantly improved thanks to advancements in novel therapeutics, including surgical resection, chemoradiotherapy, and targeted therapy [45,104,105,106]. The 5-year survival rate at the initial cancer stage is almost 90% [107]; however, many patients still experience resistance to these drugs and develop distant metastasis [108]. Thus, there is an urgent need to identify novel and specific biomarkers, which will assist in the early detection of CRC and improve strategies for drug development [9,15,109]. c-Met tyrosine kinase was reported to be a prognostic marker in primary and metastatic cancer [110]. Studies showed that c-Met plays a significant role in cancer progression in different cancers [13,14,20,111,112]; moreover, dysregulation of the c-Met pathway was demonstrated in CRC, although still little is known about its genetic mutations in CRC [110,111,113]. In addition, others showed that c-Met is expressed in approximately 60% of CRC patients, and its upregulation is associated with disease progression [113].

Accumulating studies have demonstrated that c-Met is a target gene of WNT/β-catenin signaling [114], which significantly promotes the deregulation of the protoMYC oncogene in CRC. MYC has been extensively investigated as a predominant cancer-causing gene and is associated with the cell cycle, differentiation, and apoptosis [115]. Several studies reported the overexpression of c-MYC in almost 10% of CRC patients [38,41,42]. Those findings suggest a potential treatment approach targeting c-MYC signaling. However, acquiring targeted therapies for c-MYC remains a huge challenge because MYC is located in the nucleus [46]. Previously, there were satisfactory breakthroughs in describing critical cell cycle signaling pathways in cancer initiation and progression. CCND1 was one of the identified pathways involved [116]. Studies showed that the overexpression of c-MYC enhances the expression of CCND1 through beta-catenin activation [45]. CCND1 is a crucial checkpoint protein that regulates the transition of the G1 to S phase; moreover, CCND1 was shown to promote tumorigenesis, invasion, and resistance to apoptosis [117].

In CRC, CCND1 overexpression has yielded conflicting results, as some researchers demonstrated its role in promoting poor clinical outcomes, while others reported it as a good prognostic marker [117,118,119,120]. Those findings thus suggest crosstalk among c-Met/GSK3β/MYC/CCND1 in regulating the cell cycle in CRC. Herein, we conducted a bioinformatics simulation and identified the 25 top DEGs in CRC. Interestingly, we found the c-Met/GSK3β/MYC/CCND1 signaling pathway to be expressed among these top 25 genes (Figure 2). In a further analysis, we applied a PPI network (PIN) functional enrichment analysis. We discovered that c-Met/GSK3β/MYC/CCND1 were coexpressed in the same clustering network. When we further performed an enrichment analysis, we found that enriched GO BPs involved the transcription factor network, G1 phase, cyclin D, and mitosis G1–G1/S phases (Figure 3). Recently, considerable attention has focused on small molecules as targeted therapies for cancer treatment [35]. In this study, we demonstrated the anticancer activities of our newly synthesized quercetin-derived compound, NSC772864, against panels of NCI human CRC cell lines. The compound exhibited cytotoxic activities against CRC cell lines as indicated by the percentage growth altered by treatment; the growth inhibition percentage revealed that HCC-2988 cells were more sensitive to the compound with growth inhibition of −88.65%, followed by KM12 (at −53.42%), HCT15 (at −50.38%), COLO205 (at −45.69%), SW620 (at −43.72%), HCT116 (at −36.96%), and HT29 cells (at −9.75%).

Moreover, we utilized the swissADME web platform and found that NSC772864 passed the drug-likeness, PK, medicinal chemistry, and ADMET criteria, as shown in Figure 5. We further used the DTP-COMPARE platform and PharmMapper software and predicted the c-Met, c-MYC, and CCND1 signatures as target genes of NSC772864 (Figure 16). To validate these results, we evaluated protein–ligand interactions through a molecular docking analysis. The results obtained from the ligand–receptor complexes show unique, different binding energies of NSC772864 with c-Met (ΔG = −8.0 kcal/mol), GSK3β (ΔG = −9.1 kcal/mol), MYC (ΔG = −9.1 kcal/mol), and CCND1 (ΔG = −8.0 kcal/mol). These results were further compared to standard inhibitors of crizotinib for c-Met, AZD1080 for GSK3β, alobresib for MYC, and trilacoclib for CCND1, with respective binding energies of ΔG = −8.1, −8.6, −7.6, and −7.4 kcal/mol. These results suggest that NSC772864 is a potential and novel compound for treating c-Met/GSK3β/MYC/CCND1 in CRC. Collectively, these findings create an opportunity for a novel research approach for NSC772864 as a novel and potential antiCRC agent.

## 5. Conclusions

We identified c-Met/GSK3β/MYC/CCND1 oncogenic signature as major players in CRC progression through the cell cycle. Our bioinformatics analysis demonstrated that these oncogenic signatures are overexpressed in CRC tissues compared to adjacent normal tissues, resulting in tumor progression, metastasis, poor prognoses, and resistance to current chemotherapies and targeted therapies. We further demonstrated the anticancer activities of our newly synthesized quercetin-derived small molecule, NSC772864, against NCI human CRC cell lines and its dose-dependent cytotoxic preference for CRC. Our target prediction analysis showed that c-Met/GSK3β/MYC/CCND1 signatures are target genes for NSC772864, and these findings were validated thought a molecular docking analysis, which showed remarkable binding energies of NSC772864 in a complex with c-Met, GSK3β, c-MYC, and CCND1. These findings create an opportunity for a novel research approach for NSC772864 as a novel and potential antiCRC agent. Further in vitro and in vivo investigations to validate the antiproliferative and cytotoxic activities of NSC772864 are ongoing.

## Figures and Tables

**Figure 1 cells-12-00340-f001:**
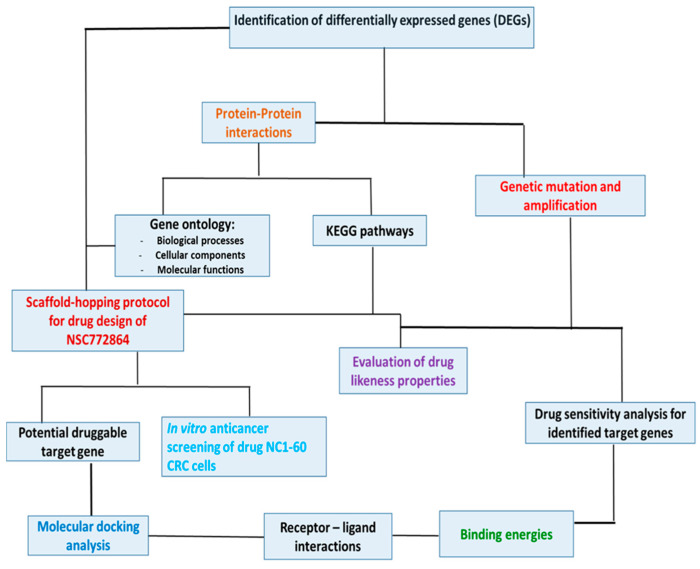
Schematic diagram representing the study design.

**Figure 2 cells-12-00340-f002:**
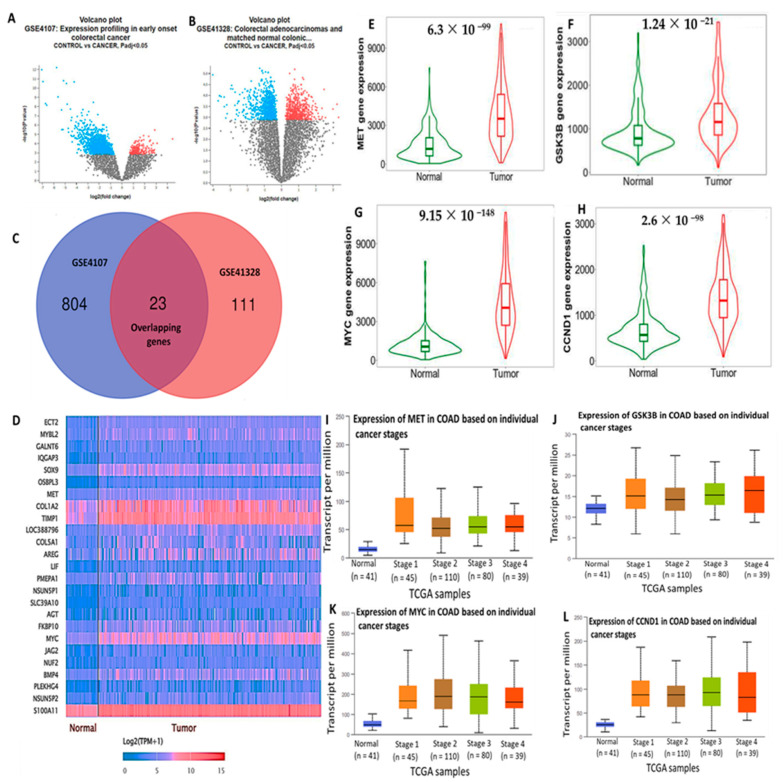
Overexpression of *c-Met/GSK3β/MYC/CCND1* promotes colon cancer progression. (**A**,**B**) Volcano plots showing top overexpressed DEGs in COAD tissues compared to normal tissues were retrieved from two NCBI GEO datasets, GSE4107 and GSE41328. (**C**) Venn diagram showing the overlapping upregulated genes from the two datasets. (**D**) Heatmaps showing the top differentially expressed genes in colon adenocarcinoma (COAD). (**E**–**H**) c-Met, GSK3β, MYC, and CCND1 were highly expressed in colon cancer tissues compared to adjacent normal tissues. (**I**–**L**) *c-Met/GSK3β/MYC/CCND1* were overexpressed in colon cancer from stages 1~4 compared to normal tissues, with *p* < 0.05. considered statistically significant.

**Figure 3 cells-12-00340-f003:**
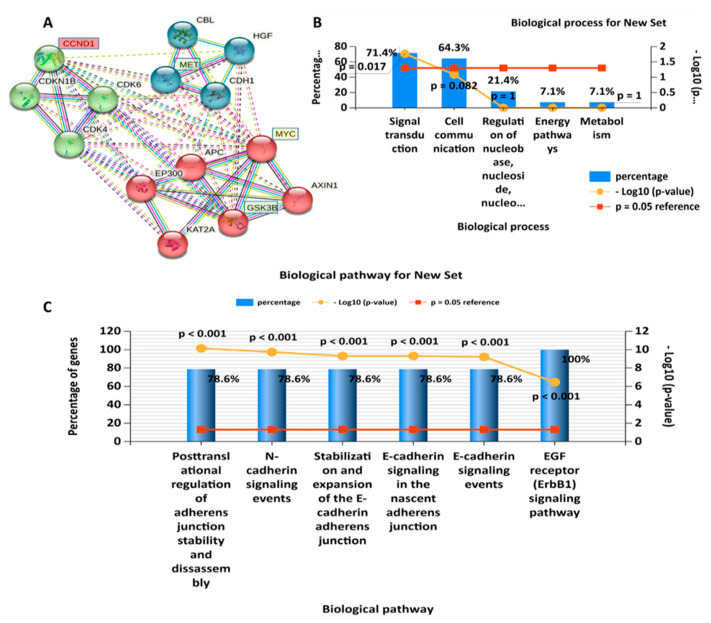
Protein–protein interaction (PPI) network revealed interactions among the *c*-Met/GSK3β/MYC/CCND1 oncogenes in colorectal cancer (CRC). (**A**) The average local clustering coefficient was 0.920 with interaction enrichment of *p* = 0.0135. Interactions were based on coexpressions, gene fusion, and co-occurrences. (**B**) The top 10 biological processes (BPs), (**C**) KEGG pathways, with *p* < 0.05 were considered significant.

**Figure 4 cells-12-00340-f004:**
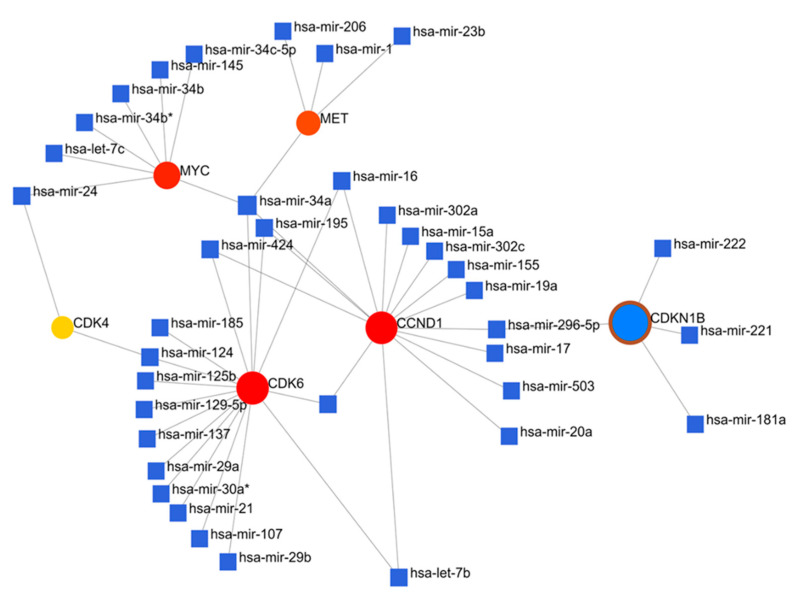
Network topology profiling of miRNAs that targets c-Met, GSK3β, MYC, and CCND1 oncogenes specifically, with * *p* < 0.05 considered statistically significant.

**Figure 5 cells-12-00340-f005:**
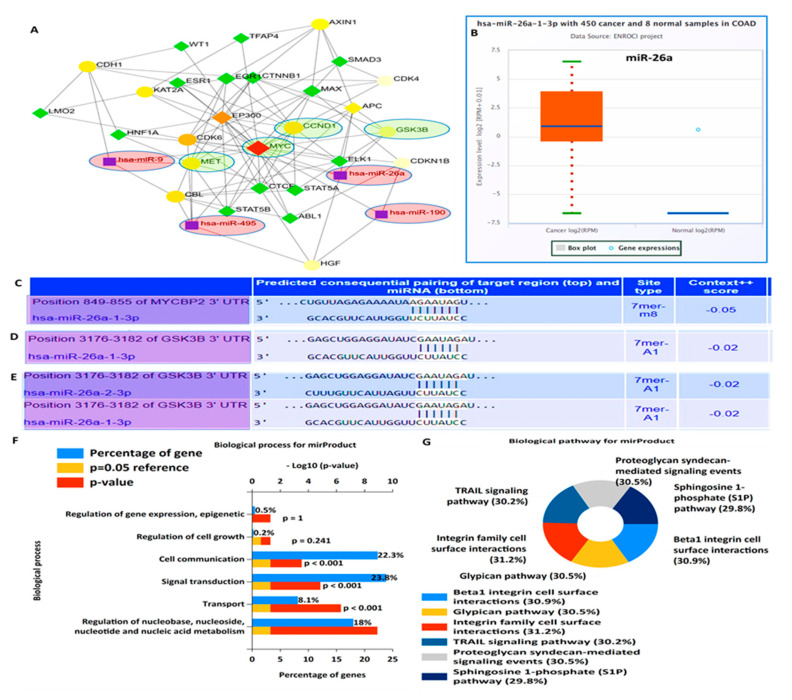
Crosstalk between overexpression of Met/GSK3β/MYC/CCND1 oncogenes and upregulated miR-26a is associated with CRC cancer progression. (**A**) expressed miRNA interacting with Met/GSK3β/MYC/CCND1 within the same cluster. (**B**) miRNA-26a is significantly downregulated in CRC samples compared to normal samples. (**C**–**E**) miR-26a-1-3p site in the 3′ untranslated region (UTR) of GSK3β, MYC, and CCND1 mRNA, thus predicting a link between miR-26a and Met/GSK3β/MYC/CCND1 in cancers. (**F**–**G**) Enriched GO, including biological processes (BPs), and affected KEGG pathways, with *p* < 0.05 considered statistically significant.

**Figure 6 cells-12-00340-f006:**
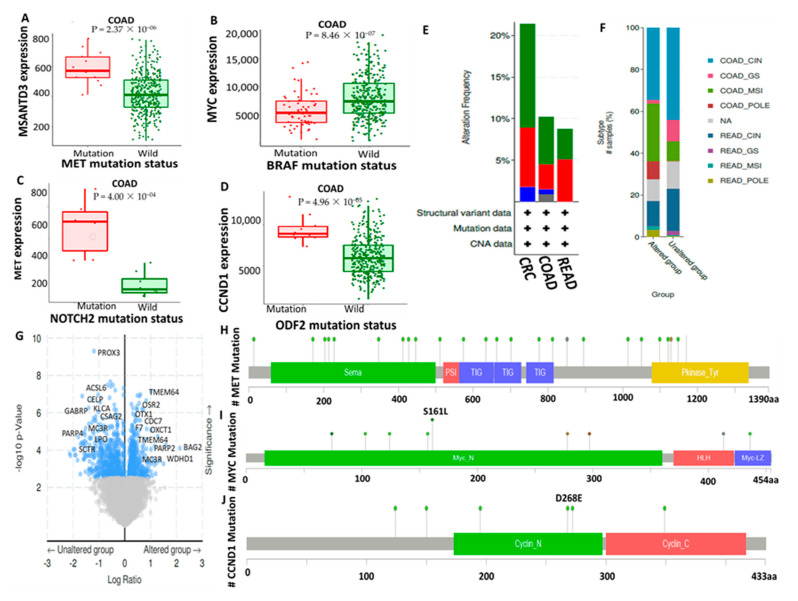
Genomic alterations in c-Met/MYC/cyclin D1 (CCND1) signatures are associated with poor prognoses of colorectal cancer (CRC) cohorts. (**A**) Mutation of the c-Met gene was associated with high expression levels of MSANTD3. (**B**–**D**) Changes in expression levels of the c-Met/GSK3β/MYC/CCND1 oncogenes were linked to mutations of the BRAF, NOTCH2, and ODF2 genes at the target level, compared to the wild-type (WT) group. (**E**) Bar graph showing higher alteration frequencies of c-Met/GSK3β/MYC/CCND1 gene signatures in CRC, compared to colorectal adenoma (COAD) and rectal adenocarcinoma (READ). (**F**) Bar graph showing alterations of the c-Met/GSK3β/MYC/CCND1 genes according to sample subtypes. (**G**) Volcano plot showing unaltered and altered genes associated with c-Met/GSK3β/MYC/CCND1 signatures with −log10(*p* values) set as a standard. (**H**–**J**) A lollipop diagram showing the distribution of c-Met/GSK3β/MYC/CCND1 mutations in CRC across protein domains. Mutations are color-coded as missense, truncating, and in-frame mutations.

**Figure 7 cells-12-00340-f007:**
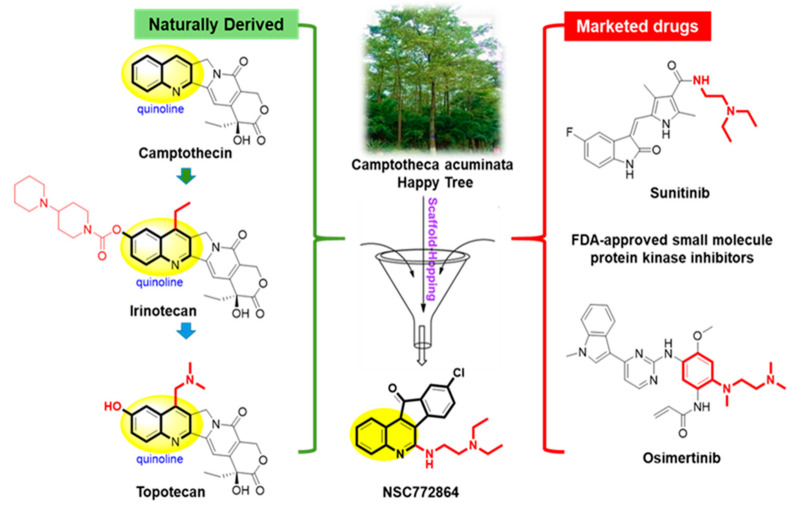
Rational scaffold-hopping protocol for the design of NSC772684.

**Figure 8 cells-12-00340-f008:**
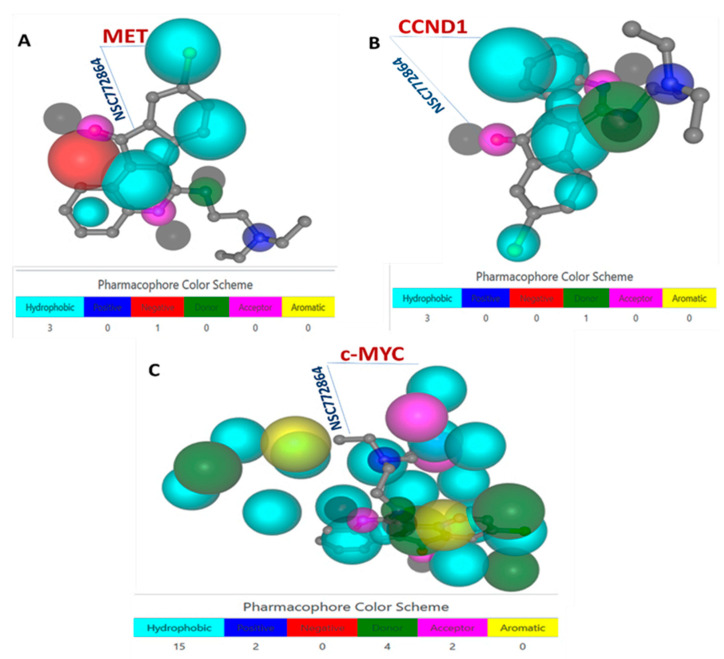
Identified potential targets for the NSC772864 small molecule. (**A**–**C**) Pharmacophore-based models of c-Met/cMYC/CCND1 identified as NSC772864 target genes through protein–receptor interactions.

**Figure 9 cells-12-00340-f009:**
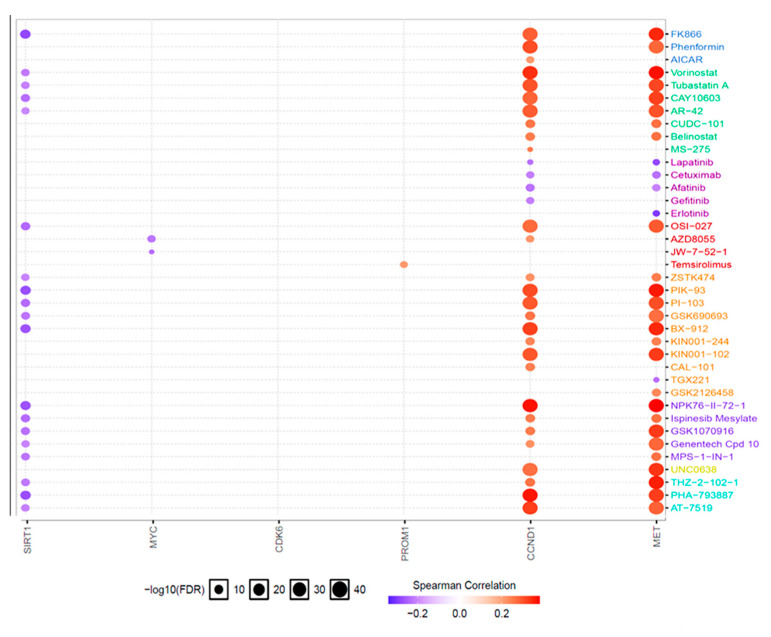
Drug sensitivity of the c-Met/GSK3β/MYC/CCND1 oncogenes from the GSCA. Correlations among genomics of drug sensitivity in cancer (GDSC) of FDA-approved drugs. Positive Spearman correlation coefficients (orange circles) indicate that increased gene expression levels were resistant to the drug, compared to negative correlations shown in blue, which indicate sensitivity to the drug.

**Figure 10 cells-12-00340-f010:**
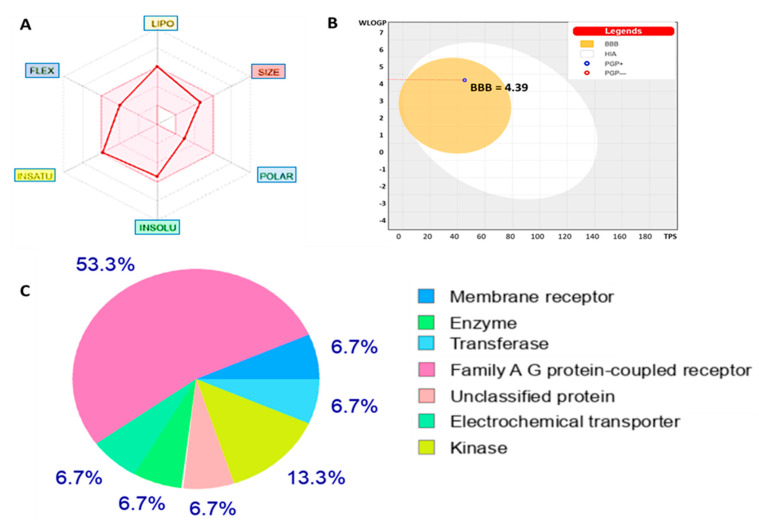
NSC772864 passed the required physicochemical properties, medicinal chemistry, pharmacokinetics (PKs), and drug-likeness criteria. (**A**) Structure of the NSC772864 small molecule, bioavailability (BA) radar, displaying the six physicochemical properties of absorption including lipophilicity (XLOGP3 = 5.04), molecular weight (379.88 g/mol), polarity (PSA = 45.23 Å^2^), solubility (Log S (ESOL) = −5.41), flexibility (rotation = 6), saturation (fraction Csp3 = 0.27), and pKa of the most basic or acidic group (=0.6) of the NSC772864 compound. In addition, the NSC772864 compound demonstrated a highly probable gastrointestinal absorption (GIA), BA score (55%), and good synthetic accessibility (3.26). (**B**) The compound passed the blood–brain barrier (BBB) assessment with a score of 5.98 and further displayed a drug-like model score of −0.48. A structural characterization of the compound was performed with the help of spectroscopic studies including IR, proton NMR, 13C NMR, MS, and an elemental analysis (**C**). Pie chart showing targetable protein candidates for NSC772864 (Table 2). Results are shown of the physiochemical properties, pharmacokinetics, drug-likeness, and medical chemistry.

**Figure 11 cells-12-00340-f011:**
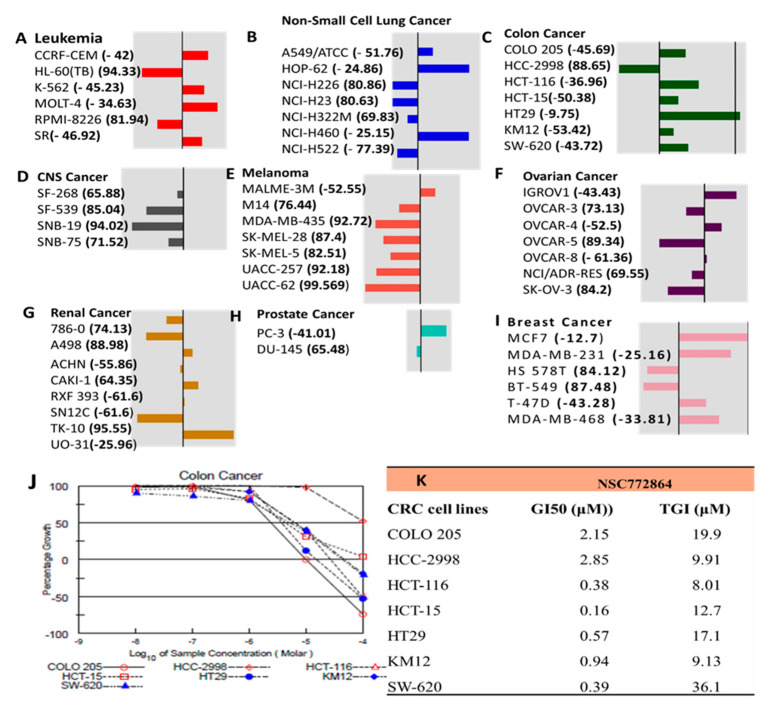
NSC772864 demonstrated anticancer effects in NC-I60 human colon cancer cell lines. (**A**–**I**) Single-dose treatment administered at 10 µM revealed growth percentage antiproliferative effects against COLO205, HCC-2998, HCT-116, HCT-15, KM12, and SW620 cells and cytotoxic effects against the HT29 cell line. (**J**) Dose-dependent responses of colon cancer cell lines evaluated using the 50% growth inhibition (GI50) and tumor growth inhibition (TGI). (**K**) In vitro IG50 results ranged from 0.16 to 2.85 μM in colon cancer cell lines, with HCT-15 being the most responsive at 0.16 μM, followed by HCT116 at 0.38 μM, SW-620 at 0.39 μM, HT-29 at 0.57 μM, KW12 at 0.94 μM, and COLO 205 at 2.15 μM, with HCC-2998 at 2.85 μM showing the least responsiveness compared to the aforementioned cell lines.

**Figure 12 cells-12-00340-f012:**
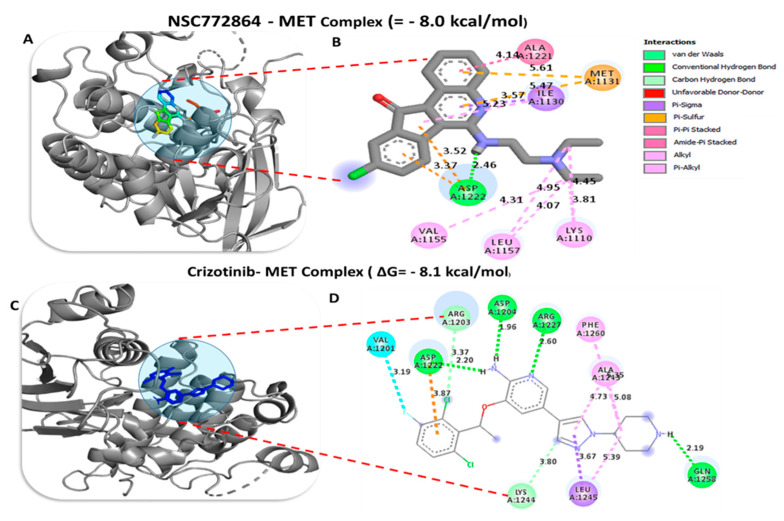
Docking profiles of c-Met with NSC772864 and standard inhibitors. (**A**,**B**) NSC772864 bound with c-Met with Gibbs’ free energy (ΔG) of −8.0 kcal/mol. (**C**,**D**) Binding of crizotinib with c-Met showed a slightly higher affinity of ΔG = −8.1 kcal/mol compared to NSC772864. Table 3 gives binding energies of ligand–receptor interactions, including different types of interactions and the amino acid residues involved.

**Figure 13 cells-12-00340-f013:**
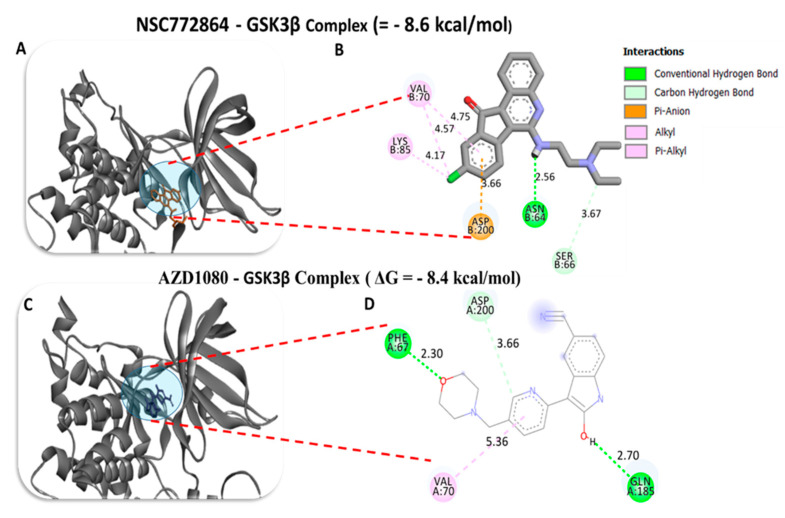
Docking profiles of GSK3β with NSC772864 and standard inhibitors. (**A**,**B**) NSC772864 bound with GSK3β with Gibbs’ free energy (ΔG) of −8.6 kcal/mol. (**C**,**D**) Binding of AZD1080 with GSK3β showed a slightly higher affinity of ΔG = −8.4 kcal/mol compared to NSC772864. Table 4 gives binding energies of ligand–receptor interactions, including different types of interactions and the amino acid residues involved.

**Figure 14 cells-12-00340-f014:**
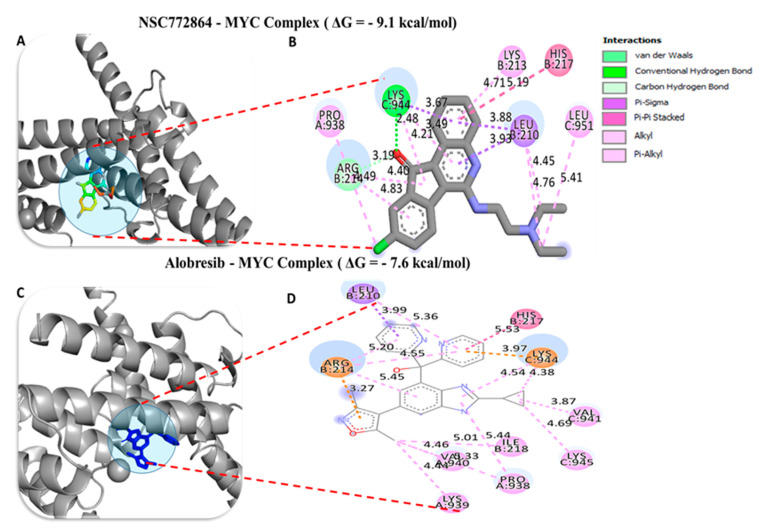
Docking profiles of c-MYC with NSC772864 and standard inhibitors. (**A**,**B**) NSC772864 bound with c-MYC with Gibbs’ free energy (ΔG) of −9.1 kcal/mol. (**C**,**D**) Binding of crizotinib with c-Met showed a slightly higher affinity of ΔG = −7.6 kcal/mol compared to NSC772864. Table 5 gives binding energies of ligand–receptor interactions, including different types of interactions and the amino acid residues involved.

**Figure 15 cells-12-00340-f015:**
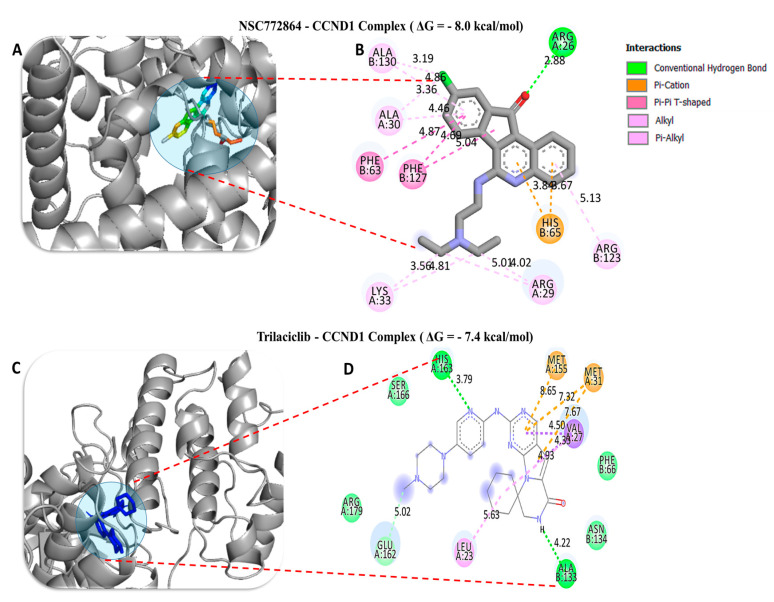
Docking profiles of cyclin D1 (CCND1) with NSC772864 and standard inhibitors. (**A**,**B**) NSC772864 bound to CCND1 with Gibbs’ free energy (ΔG) of −8.0 kcal/mol. (**C**,**D**) Binding of trilaciclib with c-Met showed a slightly higher affinity at ΔG = −7.4 kcal/mol compared to NSC772864. Table 6 gives the binding energies of ligand–receptor interactions, including different types of interactions and the amino acid residues involved.

**Figure 16 cells-12-00340-f016:**
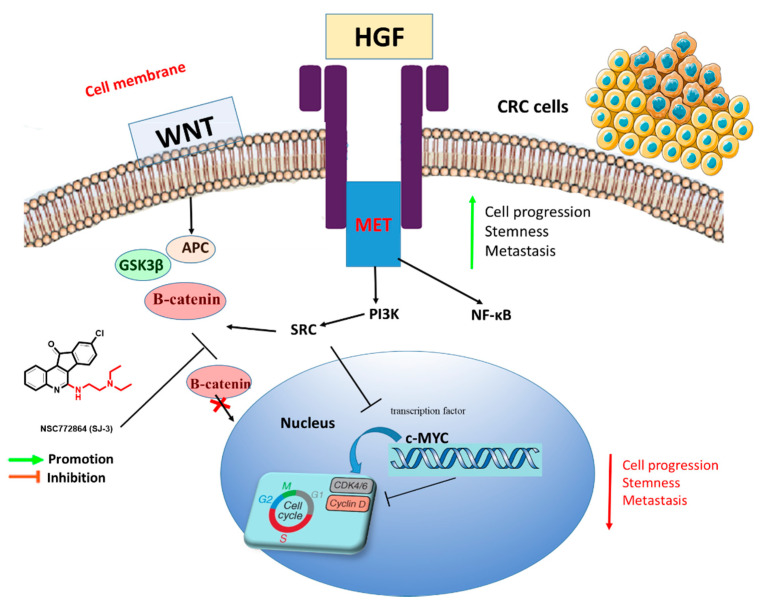
Schematic diagram shows NSC772864 may directly target Met receptor-mediated cell cycle progression through WNT/β-catenin/GSK3β pathway and MYC transcription factor. c-Met expression enhances the Wnt/β-catenin signaling and impedes GSK3β from phosphorylating β-catenin in CRC; this subsequently promotes β-catenin translocation into the nucleus leading to cancer initiation; moreover, proto-oncogene, c-MYC, is associated with the regulation of CCND1, a key regulatory protein through the G1 to S phases of the cell cycle via binding to cyclin-dependent kinase 4 (CDK4) and CDK6 cell cycle, and play crucial roles in cell cycle progression. Therefore, blockade of c-Met/cMYC/CCND1 inhibits CRC cell progression, stemness, and metastasis.

**Table 1 cells-12-00340-t001:** Correlation of SJ3 with NCI synthetic compounds and standard anticancer agents sharing similar anticancer fingerprints and mechanisms.

		NCI Synthetic Compounds			NCI Standard Agents			ArrayCGH-Gray	
Rank	r	CCLC	Target Descriptor	r	CCLC	Target Descriptor	r	CCLC	Target Descriptor
1	0.61	56	Raloxifene	0.46	55	Tamoxifen	0.27	53	I.G.F.A.
2	0.52	55	Majoranolide	0.37	46	Menogaril	0.23	57	c-Met
3	0.51	57	Tyloxapol (usan)	0.36	43	Mitramycin	0.12	55	CCND1
4	0.53	52	Tolonium chloride	0.28	56	Tamoxifen	0.21	53	WNT1
5	0.54	51	Ivosidenib	0.28	55	Fluorodopan	0.26	55	CDK6
6	0.56	56	Bafilomycin deriv	0.42	59	Thioguanine	0.16	51	AKT2
7	0.43	53	Asbestinin-d	0.31	56	Amonafide	0.14	55	T.G.F.A.
8	0.61	48	Ml148	0.13	51	Tetraplatin	0.13	54	MYC
9	0.56	52	Raloxifene	0.3	58	Rapamycin	0.3	53	MMP8
10	0.58	55	Azd-1390	0.41	56	Actinomycin D	0.1	54	PIK3CA

**Table 2 cells-12-00340-t002:** Physiochemical properties, pharmacokinetics, drug-likeness, and medical chemistry.

Physicochemical Properties Based on Bioavailability Radar of NSC765600		Recommended Value	Pharmacokinetics		
			GI Absorption	High	
Formula	C_21_H_17_F_2_NO_4_		BBB	Yes (4.39)	
Molecular weight	379.88 g/mol	150–500 g/mol	**Drug-likeness**		
Fraction Csp3	0.27	≤1	Lipinski	Yes; 0 violation	
Num. rotatable bonds	6	≤10	Ghose	Yes	
Num. H-bond acceptors	3	≤12	Veber	Yes	
Num. H-bond donors	1	≤5	Egan	Yes	
Molar Refractivity	111.86		Muegge	Yes	
TPSA	45.23 Å^2^	≤140 Å^2^	Bioavailability Score	0.55 (55%)	
Log P_o/w_ (XLOGP3)	5.04	−5.7	**Medical Chemistry**		
Log S (ESOL)	−5.41	0–6	Synthetic accessibility	3.26	1 (easy to make) and 10 (difficult to make)

**Table 3 cells-12-00340-t003:** NSC772864 and standard drug comparative docking profiles against c-Met.

NSC772864–c-Met Complex (ΔG = −8.0 kcal/mol)		Crizotinib–c-Met Complex (ΔG = −8.1 kcal/mol)	
*Type of interactions and number of bonds*	*Distance of interacting amino acids*	*Type of interactions and number of bonds*	*Distance of interacting amino acids*
Conventional Hydrogen bond (3)	ASP1222 (2.20 Å), GLU1127 (1.85 Å), and LYS1110 (2.68 Å)	Conventional Hydrogen bond (3)	ASP1222 (2.20 Å), GLU1258 (2.19 Å), ASP1204 (1.96 Å) and ARG1227 (2.60 Å)
Van der Waals forces	VAL1155, GLY1128, LYS1161, GLY1163, GLY1085, LEU1140	Carbon–Hydrogen bond	LYS1244
Carbon–Hydrogen bond	PRO1158, c-Met1160	Pi–Sigma	LEU1245
Pi–Sigma	ILE1084, LEU1157, c-Met1211		
Pi–Sulfur	c-Met1131	Alkyl	PHE1260
Amide–Pi stacked	TYR1159	Pi–Alkyl	ALA1243
Pi–Pi stacked	ALA1228		
Alkyl	VAL1092, ALA1108		
Pi–Alkyl	PHE1223		

**Table 4 cells-12-00340-t004:** NSC772864 and standard drug comparative docking profiles against GSK3β.

NSC772864–GSK3β Complex (=−8.6 kcal/mol)		AZD1080–GSK3β Complex (ΔG = −8.4 kcal/mol)	
*Type of interactions and number of bonds*	*Distance of interacting amino acids*	*Type of interactions and number of bonds*	*Distance of interacting amino acids*
Conventional Hydrogen bond (1)	ASN64 (2.56 Å)	Conventional Hydrogen bond (2)	PHE67(2.30Å), GLN185 (2.70 Å)
Carbon–Hydrogen bond	SER66	Carbon–Hydrogen bond	ASP200
Pi–Anion	ASP200	Pi–Sigma	VAL70
Pi–Alkyl	VAL70		
Alkyl	LYS85		

**Table 5 cells-12-00340-t005:** NSC772864 and standard drug comparative docking profile against c-MYC.

NSC772864–MYC Complex (ΔG = −9.1 kcal/mol)		Alobresib–MYC Complex (ΔG = −7.6 kcal/mol)	
*Type of interactions and number of bonds*	*distance of interacting Amino acids*	*Type of interactions and number of bonds*	*distance of interacting Amino acids*
Conventional Hydrogen bond (1)	SER221 (2.48 Å)	Pi-cation	ARG214, LYS944
Van der Waals forces	SER221, THR947, GLU221, SER224, ASP220	Pi-sigma	LEU210
Carbon hydrogen bond	ARG21	Pi-Pi stacked	HIS217
Pi-sigma	LEU210	Alkyl	VAL940, ILE218, LYS939
Pi-Alkyl	PRO938, LYS213, LEU651	Pi-alkyl	VAL941, LYS945
Pi-Pi stacked	HIS217		

**Table 6 cells-12-00340-t006:** NSC772864 and standard drug comparative docking profiles against c-MYC.

NSC772864–CCND1 Complex (ΔG = −8.0 kcal/mol)		Trilaciclib–CCND1 Complex (ΔG = −7.4 kcal/mol)	
*Type of interactions and number of bonds*	*Distance of interacting amino acids*	*Type of interactions and number of bonds*	*Distance of interacting amino acids*
Conventional Hydrogen bond (1)	ARG26 (2.88 Å)	Conventional Hydrogen bond (2)	HIS163 (2.29 Å), ALA133 (2.04 Å)
Van der Waals forces	ASN131	Carbon–Hydrogen bond	GLU162
Pi–Cation	HIS65	Pi–Sigma	VAL27
Pi-Pi stacked	PHE63, PHE127	Pi–Sulfur	c-Met155, c-Met31,
Alkyl	ALA130, ALA30	Pi–Alkyl	LEU23

## Data Availability

Data supporting the results of this manuscript will be made available without undue reservation.

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
