# Peer review of "Multiomics Study of a Novel Naturally Derived Small Molecule, NSC772864, as a Potential Inhibitor of Proto-Oncogenes Regulating Cell Cycle Progression in Colorectal Cancer"

_cells, 2023, doi:10.3390/cells12020340_

Round 1

Reviewer 1 Report

The study purposed to investigate the expression of c-Met/GSK3β/MYC/CCND1 by CRC cells, since these regulate various cell functions and are aberrantly expressed in certain cancers. 

Secondarily, the purpose of the study was to investigate the effects of a novel quinolone derivative against CRC cell lines, and further determine the mechanism of action using computational modelling. 

The topic addressed by the study is relevant and of interest, indeed there is a continuing quest to identify more effective anti-cancer compounds which are selective and have less side-effects, and this is what this study seeks to contribute to. The manuscript is written well, the methods are mostly appropriate to address the study objectives, the points that need attention concerning the methodology are indicated below.

- The authors should briefly state the methods employed in screening.

- A positive control should be included.

- Stats analysis for correlations of the expression of c-Met/GSK3β/MYC/CCND1 etc cells was undertaken, however, nothing is stated for other quantitative experiments.

- What was the sample size/ Any repeats?  

- What vehicle was used?- The authors need to clarify some aspects of the study so as to support the conclusions reached. The initial assertion is that current anti-cancer treatments are ‘limited’ and toxic. However, the current study does not show that the tested compound is selective (or is the compound cytotoxic to all dividing cells regardless of whether they are neoplastic). 

- The problem/limitation with most anti-cancer treatments is a lack of selectivity. 

- Data showing the effectiveness of the test compound vs a positive control should be included.The figures are clear. 

- Fig 15. Kindly change the following: ‘cell nucleus’ should be ‘nucleus’

‘cytomembrane’ should be ‘cell membrane’. The figure caption should not contain information about a table.

- What is ‘cell progression’ in Fig 15? – kindly use a more appropriate label.

- Fig 15 should be explained fully in the text. 

Reviewer 2 Report

  • Figure 3B contains a mistake, please fix this error.
  • The manuscript is based on bioinformatic results but I think that these data must be confirmed by some molecular experiments such as qPCR analyses on CRC cells treated with NSC772864 or western blotting.
  • The authors identified the overexpression of c-Met/GSK3β/MYC/CCND1 oncogenic signatures that were associated with cancer progression, drug resistance, metastasis, and poor clinical outcomes in C.R.C. As demonstrated by Delle Cave et al, L1CAM and CXCR4 are also overexpressed in CRC and their expression is correlated with some of these oncogenic pathways and their expression is predictive of poor prognosis. The authors should analyze the expression of these two markers in CRC cells treated with NSC772864 (by qPCR and/or western blotting) (DOI: 10.7150/thno.54027).

Round 2

Reviewer 2 Report

Accepted